# Adaptive Behavior Cloning Regularization for Stable Offline-to-Online Reinforcement Learning

## Abstract

Offline reinforcement learning, by learning from a fixed dataset, makes it possible to learn agent behaviors without interacting with the environment. However, depending on the quality of the offline dataset, such pre-trained agents may have limited performance and would further need to be fine-tuned online by interacting with the environment. During online fine-tuning, the performance of the pre-trained agent may collapse quickly due to the sudden distribution shift from offline to online data. While constraints enforced by offline RL methods such as a behaviour cloning loss prevent this to an extent, these constraints also significantly slow down online fine-tuning by forcing the agent to stay close to the behavior policy. We propose to adaptively weigh the behavior cloning loss during online fine-tuning based on the agent's performance and training stability. Moreover, we use a randomized ensemble of Q functions to further increase the sample efficiency of online fine-tuning by performing a large number of learning updates. Experiments show that the proposed method yields state-of-the-art offline-to-online reinforcement learning performance on the popular D4RL benchmark.

## 1 Introduction

Offline or batch reinforcement learning (RL) deals with the training of RL agents from fixed datasets generated by possibly unknown behavior policies, without any interactions with the environment. This is important in problems like robotics, autonomous driving, and healthcare where data collection can be expensive or dangerous. Offline RL has been challenging for model-free RL methods due to extrapolation error where the Q networks predict unrealistic values upon evaluations on out-of-distribution state-action pairs (Fujimoto et al., 2019). Recent methods overcome this issue by constraining the policy to stay close to the behavior policy that generated the offline data distribution (Fujimoto et al., 2019; Kumar et al., 2020; Kostrikov et al., 2021; Fujimoto & Gu, 2021), to demonstrate even better performance than the behavior policy on several simulated and real-world tasks (Siegel et al., 2020; Singh et al., 2020; Nair et al., 2020).

However, the performance of pre-trained policies will be limited by the quality of the offline dataset and it is often necessary or desirable to fine-tune them by interacting with the environment. Also, offline-to-online learning reduces the risks in online interaction as the offline pre-training results in reasonable policies that could be tested before deployment. In practice, offline RL methods often fail during online fine-tuning by interacting with the environment. This offline-to-online RL setting is challenging due to: (i) the sudden distribution shift from offline data to online data. This could lead to severe bootstrapping errors which completely distorts the pre-trained policy leading to a sudden performance drop from the very beginning of online fine-tuning, and (ii) constraints enforced by offline RL methods on the policy to stay close to the behavior policy. While these constraints help in dealing with the sudden distribution shift they significantly slow down online fine-tuning from newly collected samples.

We propose to adaptively weigh the offline RL constraints such as behavior cloning loss during online fine-tuning. This could prevent sudden performance collapses due to the distribution shift while also enabling sample-efficient learning from the newly collected samples. We propose to perform this adaptive weighing according to the agent's performance and the training stability. We

start with TD3+BC, a simple offline RL algorithm recently proposed by Fujimoto & Gu (2021) which combines TD3 (Fujimoto et al., 2018) with a simple behavior cloning loss, weighted by an $\alpha$ hyperparameter. We adaptively weigh this $\alpha$ hyperparameter using a control mechanism similar to the proportional–derivative (PD) controller. The $\alpha$ value is decided based on two components: the difference between the current return and the target return (proportional term) as well as the change of return between current episode and the last episode (derivative term). We demonstrate that these simple modifications lead to stable online fine-tuning after offline pre-training on datasets of different quality. We also use a randomized ensemble of Q functions (Chen et al., 2021) to further improve the sample-efficiency. We attain state-of-the-art online fine-tuning performance on locomotion tasks from the popular D4RL benchmark.

## 2 RELATED WORK

**Offline RL**. Offline RL aims to learn a policy from pre-collected fixed datasets without interacting with the environment (Lange et al., 2012; Agarwal et al., 2020; Fujimoto et al., 2019; Kumar et al., 2019; Nachum et al., 2019; Siegel et al., 2020; Levine et al., 2020; Peng et al., 2019). Off-policy RL algorithms allow for reuse of off-policy data (Konda & Tsitsiklis, 2000; Degris et al., 2012; Haarnoja et al., 2018; Silver et al., 2014; Lillicrap et al., 2015; Fujimoto et al., 2018; Mnih et al., 2015) but they typically fail when trained offline on a fixed dataset, even if it's collected by a policy trained using the same algorithm (Fujimoto et al., 2019; Kumar et al., 2019). In actor-critic methods, this is due to extrapolation error of the critic network on out-of-distribution state-action pairs Levine et al. (2020). Offline RL methods deal with this by constraining the policy to stay close to the behavioral policy that collected the offline dataset. BRAC (Wu et al., 2019) achieves this by minimizing the Kullback-Leibler divergence between the behavior policy and the learned policy. BEAR (Kumar et al., 2019) minimizes the MMD distance between the two policies. TD3+BC (Fujimoto & Gu, 2021) proposes a simple yet efficient offline RL algorithm by adding an additional behavior cloning loss to the actor update. Another class of offline RL methods learns conservative Q functions, which prevents the policy network from exploiting out-of-distribution actions and forces them to stay close to the behavior policy. CQL (Kumar et al., 2020) changes the critic objective to also minimize the Q function on unseen actions. Fisher-BRC (Kostrikov et al., 2021) achieves conservative Q learning by constraining the gradient of the Q function on unseen data. Model-based offline RL methods (Yu et al., 2020; Kidambi et al., 2020) train policies based on the data generated by ensembles of dynamics models learned from offline data, while constraining the policy to stay within samples where the dynamics model is certain. In this paper, we focus on offline-to-online RL with the goal of stable and sample-efficient online fine-tuning from policies pre-trained on offline datasets of different quality.

**Offline pre-training in RL**. Pre-training has been vastly investigated in the machine learning community from computer vision (Sharif Razavian et al., 2014; Donahue et al., 2014; Yosinski et al., 2014) to natural language processing (Devlin et al., 2018; Turian et al., 2010). Offline pre-training in RL could enable deployment of RL methods in domains where data collection can be expensive or dangerous. (Silver et al., 2016; Gupta et al., 2019; Rajeswaran et al., 2017) pre-train the policy network with imitation learning to speed up RL. QT-opt (Kalashnikov et al., 2018) studies vision-based object manipulation using a diverse and large dataset collected by seven robots over several months and fine-tune the policy with 27K samples of online data. However, these methods pre-train using diverse, large, or expert datasets and it is also important to investigate the possibility of pre-training from offline datasets of different quality. Yang & Nachum (2021); Ajay et al. (2020) use offline pre-training to accelerate downstream tasks. AWAC (Nair et al., 2020) and Balanced Replay Lee et al. (2021) are recent works that also focus on offline-to-online RL from datasets of different quality. AWAC updates the policy network such that it is constrained during offline training while not too conservative during fine-tuning. Balanced Replay trains an additional neural network to prioritize samples in order to effectively use new data as well as near-on-policy samples in the offline dataset. We compare with AWAC and Balanced Replay to attain state-of-the-art offline-to-online RL performance on the popular D4RL benchmark.

**Ensembles in RL**. Ensemble methods are widely used for better performance in RL (Faußer & Schwenker, 2015; Osband et al., 2016; Chua et al., 2018; Janner et al., 2019). In model-based RL, PETS (Chua et al., 2018) and MBPO (Janner et al., 2019) use probabilistic ensembles to effectively model the dynamics of the environment. In model-free RL, ensembles of Q functions have been

shown to improve performance (Anschel et al., 2017; Lan et al., 2020). REDQ (Chen et al., 2021) learns a randomized ensemble of Q functions to achieve similar sample efficiency as model-based methods without learning a dynamic model. We utilize REDQ in this work for improved sample-efficiency during online fine-tuning. Specific to offline RL, REM (Agarwal et al., 2020) uses random convex combinations of multiple Q-value estimates to calculate the Q targets for effective offline RL on Atari games. MOPO (Yu et al., 2020) uses probabilistic ensembles from PETS to learn policies from offline data using uncertainty estimates based on model disagreement. MBOP (Argenson & Dulac-Arnold, 2020) uses ensembles of dynamic models, Q functions, and policy networks to get better performance on locomotion tasks. Balanced Replay (Lee et al., 2021) uses ensembles of pessimistic Q functions to mitigate instability caused by distribution shift in offline-to-online RL. While ensembling of Q functions has been studied by several prior works (Lan et al., 2020; Chen et al., 2021), we combine it with behavioral cloning loss for the purpose of robust and sample-efficient offline-to-online RL.

**Adaptive balancing of multiple objectives in RL**. Ball et al. (2020) train policies using learned dynamics models with the objective of visiting states that most likely lead to subsequent improvement in the dynamics model, using active online learning. They adaptively weigh the maximization of cumulative rewards and minimization of model uncertainty using an online learning mechanism based on exponential weights algorithm. In this paper, we focus on offline-to-online RL using model-free methods and propose to adaptively weigh the maximization of cumulative rewards and a behavioral cloning loss. Exploration of other online learning algorithms such as exponential weights algorithm is a line of future work.

## 3 BACKGROUND

### 3.1 REINFORCEMENT LEARNING

Reinforcement learning (RL) deals with sequential decision making to maximize cumulative rewards. RL problems are often formalized as Markov decision processes (MDPs). An MDP consists of a set of states $\mathcal{S}$, a set of actions $\mathcal{A}$, a transition dynamics $\boldsymbol{s}_{t+1} \sim p(\cdot|\boldsymbol{s}_t, \boldsymbol{a}_t)$ that represents the probability of transitioning to a state $\boldsymbol{s}_{t+1}$ by taking action $\boldsymbol{a}_t$ in state $\boldsymbol{s}_t$ at timestep $t$, a scalar reward function $r_t = R(\boldsymbol{s}_t, \boldsymbol{a}_t)$, and a discount factor $\gamma \in [0, 1]$.

A policy function $\pi$ of an RL agent is a mapping from states to actions and defines the behavior of the agent. The value function $V_\pi(\boldsymbol{s})$ of a policy $\pi$ is defined as the expected cumulative rewards from state $s$: $V^\pi(\boldsymbol{s}) = \mathbb{E}[\sum_{t=0}^\infty \gamma^t R(\boldsymbol{s}_t, \boldsymbol{a}_t)|s_0 = \boldsymbol{s}]$, where the expectation is taken over state transitions $\boldsymbol{s}_{t+1} \sim p(\cdot|\boldsymbol{s}_t, \boldsymbol{a}_t)$ and policy function $\boldsymbol{a}_t \sim \pi(\boldsymbol{s}_t)$. Similarly, the state-action value function $Q^\pi(\boldsymbol{s}, \boldsymbol{a})$ is defined as the expected cumulative rewards after taking action $a$ in state $s$: $Q^\pi(\boldsymbol{s}, \boldsymbol{a}) = \mathbb{E}[\sum_{t=0}^\infty \gamma^t R(\boldsymbol{s}_t, \boldsymbol{a}_t)|s_0 = \boldsymbol{s}, \boldsymbol{a}_0 = \boldsymbol{a}]$. The goal of RL is to learn an optimal policy function $\pi_\theta$ with parameters $\theta$, that maximizes the expected cumulative rewards:

$$\pi_\theta = \arg\max_\theta \mathbb{E}_{\boldsymbol{s} \sim \mathcal{S}}\left[V^{\pi_\theta}(\boldsymbol{s})\right] = \arg\max_\theta \mathbb{E}_{\boldsymbol{s} \sim \mathcal{S}}\left[Q^{\pi_\theta}(\boldsymbol{s}, \pi_\theta(\boldsymbol{s}))\right].$$

We use the TD3 algorithm for reinforcement learning (Fujimoto et al., 2018). TD3 is an actor-critic method that alternatingly trains: (i) the critic network $Q_\phi$ to estimate the $Q^{\pi_\theta}(\boldsymbol{s}, \boldsymbol{a})$ values of the policy network $\pi_\theta$, and (ii) the policy network to produce actions that maximize the Q function: $\nabla_\theta Q_\phi(\boldsymbol{s}, \pi_\theta(\boldsymbol{s}))$.

### 3.2 OFFLINE PRE-TRAINING

Offline reinforcement learning or batch reinforcement learning assumes that the agent is not able to interact with the environment but is given a fixed dataset $\mathcal{D}$ of $(\boldsymbol{s}, \boldsymbol{a}, r, \boldsymbol{s}')$ tuples to learn from. The data is assumed to be collected by an unknown behavioural policy (or a collection of policies).

The problem with using actor-critic methods for offline RL is extrapolation error due to the evaluation of the critic network on the next state and next action values $Q(\boldsymbol{s}', \boldsymbol{a}')$ to compute the temporal difference error. Here the next action $\boldsymbol{a}'$ is sampled from the policy network $\boldsymbol{a}' \sim \pi_\theta(\boldsymbol{s}')$ and this could lead to out-of-distribution evaluations of the critic network. This is problematic as erroneous predictions of the critic on unfamiliar actions could be propagated to other critic predictions due to

bootstrapping in temporal difference learning. This will also lead to the policy network preferring actions with unrealistic value predictions. This problem can be overcome either by constraining the policy network to stay close to the data distribution (Fujimoto & Gu, 2021) or by enforcing conservative estimates of the critic network on out-of-distribution samples (Kumar et al., 2020).

Fujimoto & Gu (2021) propose TD3+BC, a simple offline RL algorithm that regularizes policy learning in TD3 with a behavior cloning loss that constraints the policy actions to stay close to the actions in the offline dataset $\mathcal{D}$. This is achieved by adding a behavior cloning term to the policy loss:

$$\pi_\theta = \arg\max_\theta \mathbb{E}_{(\boldsymbol{s},\boldsymbol{a})\sim\mathcal{D}}\Big[\bar{Q}(\boldsymbol{s},\pi_\theta(\boldsymbol{s})) - \alpha(\pi_\theta(\boldsymbol{s}) - \boldsymbol{a})^2\Big] \tag{1}$$

where $\alpha$ is a weighing hyperparameter and

$$\bar{Q}(\boldsymbol{s},\pi_\theta(\boldsymbol{s})) = \frac{Q(\boldsymbol{s},\pi_\theta(\boldsymbol{s}))}{\frac{1}{N}\sum_{\boldsymbol{s}_i,\boldsymbol{a}_i} Q(\boldsymbol{s}_i,\boldsymbol{a}_i)}$$

normalizes the $Q$ values which help in balancing both losses. The sum in the denominator is taken over a mini-batch and the gradients do not flow through the critic term in the denominator.

## 4 ONLINE FINE-TUNING

RL agents trained from offline data tend to have limited performance and would further need to be fine-tuned online by interacting with the environment. During online fine-tuning, the performance of the pre-trained agent may collapse quickly due to the sudden distribution shift from offline data to online data. Keeping the constrain used in offline pre-training, such as in Equation 1, could mitigate the collapse. However, this will force the policy to stay close to the behavior policy (used to collect the dataset), thus leads to slow improvement. In this section, we describe the two components of our online-tuning algorithm that enables stable and sample-efficient online fine-tuning.

### 4.1 ADAPTIVE WEIGHING OF BEHAVIOR CLONING LOSS

The most straightforward way to fine-tune the pre-trained policy is by just removing the constrains used in offline pre-training. For example, Balanced Replay (Lee et al., 2021) uses CQL (Kumar et al., 2020) during offline pre-training and uses SAC (Haarnoja et al., 2018) in fine-tuning. However, this strategy often leads to a performance collapse at the beginning of fine-tuning, as shown in Fig. 1 (with $\alpha = 0$) and the TD3-ft in Fig. 2. In the TD3+BC algorithm we consider in this paper, an $\alpha$ hyperparameter is used to balance the RL objective and the behaviour cloning term which constrains the policy to stay close to the behavior policy (see Equation 1). We use $\alpha_{\text{offline}}$ and $\alpha_{\text{online}}$ to distinguish the $\alpha$ hyperparameter value used during offline and online training respectively. By default, we use $\alpha_{\text{offline}} = 0.4$ in all our experiments. We use $\alpha_{\text{online}} = 0.4$ for TD3+BC and we observe that this prevents sudden performance drops at the initial steps of online fine-tuning, at the cost of very slow learning due to the strong behavior cloning constraint. On the other hand, setting $\alpha_{\text{online}} = 0$ leads to sample-efficient learning on some tasks at the cost of complete instability in other tasks. This is due to the sudden distribution shift causing the policy network to change significantly.

In Fig. 1, we present the influence of $\alpha_{\text{online}}$ on the TD3+BC during fine-tuning by trying different values of $\alpha_{\text{online}}$ from $[0.0, 0.1, 0.3]$. We can clearly see that using the behavior cloning loss with proper $\alpha_{\text{online}}$ enables stable fine-tuning. However, the value of $\alpha_{\text{online}}$ depends on the quality of the offline dataset and has significant influence of the fine-tuning performance. For example, $\alpha_{\text{online}} = 0$ fits well on the Hopper-Random task while causes immediate collapse on Hopper-Medium and Hopper-Medium-Expert tasks.

In our experiments, we found that when the offline dataset has narrow distribution or when the policy has already converged to a desired performance (comparable to the expert), it is usually beneficial to maintain a higher $\alpha_{\text{online}}$. When the data distribution is broader or when we still need to improve the agent by a large margin, a smaller $\alpha_{\text{online}}$ works better. During experiments, we can not find a single $\alpha_{\text{online}}$ that is suitable for all tasks and its value needs to be tuned carefully per tasks, which makes this method hard to be used in practice.

To solve this problem, we propose to adapt the weight of the behavior cloning loss according to two factors: (i) the difference between current episodic return and the target return, and (ii) the episodic

Figure 1: Results of online fine-tuning on the D4RL benchmark using TD3+BC with different $\alpha_{\text{online}}$ hyperparameters. We plot the mean and standard deviation across 3 runs. Using the behavior cloning loss with proper $\alpha_{\text{online}}$ enables the stable fine-tuning. But the optimal value of $\alpha_{\text{online}}$ differs between datasets.

return between current episode and the last episode. We adaptively change the $\alpha_{\text{online}}$ hyperparameter as:

$$\Delta(\alpha_{\text{online}}) = K_P \cdot (R_{\text{current}} - R_{\text{target}}) + K_D \cdot \max(0, R_{\text{last}} - R_{\text{current}}) \tag{2}$$

where we constrain $\alpha_{\text{online}}$ between 0 and 0.4 (the value used during offline pre-training). $R_{\text{current}}$ and $R_{\text{last}}$ are normalized following the return normalization procedure used in D4RL. $R_{\text{target}}$ is the target episodic return, which we set as 1 (corresponding to the expert policy) for all tasks. $K_P$ controls how fast we decrease the $\alpha_{\text{online}}$ according to current performance and $K_D$ determines how fast we increase the $\alpha_{\text{online}}$ when the performance drops. Intuitively, when the agent's performance reaches the target episodic return, we try to maintain it during fine-tuning. But when the agent's performance is low, we decrease the $\alpha_{\text{online}}$ to allow the agent improving further. The second term increases the $\alpha_{\text{online}}$ when performance drops during training to mitigate performance collapse. Equation 2 allows for adaptive weighing of the behavior cloning loss throughout online fine-tuning. This learning algorithm can automatically adjust the constraint enforced by the behavior cloning loss.

After offline pre-training the replay buffer is filled with offline samples and during online fine-tuning, they are slowly replaced by online samples. Uniformly sampling mini-batches from this replay buffer for online fine-tuning is inefficient as it is dominated by offline samples. After offline learning we simply remove 95% of random offline samples from the replay buffer to deal with this problem. Our results show that the data down-sampling allows efficient usage of novel data without destroying the training.

### 4.2 Randomized Ensembles of Critic Networks

We propose to use an ensemble of Q functions to better deal with the distribution shift from offline pre-training and to improve the sample-efficiency of online fine-tuning. We use the Randomized Ensembled Double Q-learning (REDQ) method proposed by Chen et al. (2021) to learn an ensemble of critic networks.

The critic network is trained to satisfy the Bellman equation: $Q^\pi(\boldsymbol{s}, \boldsymbol{a}) = r + \gamma Q^\pi(\boldsymbol{s}', \pi_\theta(\boldsymbol{s}'))$. REDQ maintains an ensemble of $N$ critic networks and randomly samples $M$ networks for each critic update. Given a mini-batch $\mathcal{B}$ of $B$ transitions $(\boldsymbol{s}, \boldsymbol{a}, r, \boldsymbol{s}')$, all critic networks in the ensemble are updated towards the same target:

$$\nabla_{\phi_i} \frac{1}{|B|} \sum_{(\boldsymbol{s}, \boldsymbol{a}, r, \boldsymbol{s}') \in \mathcal{B}} \left( Q_{\phi_i}(\boldsymbol{s}, \boldsymbol{a}) - r - \gamma \min_{i \in \mathcal{M}} Q_{\phi_i}(\boldsymbol{s}', \boldsymbol{a}') \right)^2 \tag{3}$$

where $\mathcal{M}$ is a random subset of $M$ critic networks and $\boldsymbol{a}' = \text{clip}(\pi_\theta(\boldsymbol{s}_{t+1}) + \epsilon, a_{low}, a_{high})$. Here $\epsilon$ is Gaussian exploration noise with standard deviation $\sigma_{\text{policy}}$ and $[a_{low}, a_{high}]$ is the action range.

REDQ updates the policy network to maximize the average predictions of the critic networks:

$$\nabla_\theta \frac{1}{|B|} \sum_{\boldsymbol{s} \in \mathcal{B}} \frac{1}{N} \sum_{i=1}^{N} Q_{\phi_i}(\boldsymbol{s}, \pi_\theta(\boldsymbol{s})).$$

---

**Algorithm 1** Offline-to-online RL with adaptive behaviour cloning and ensembles of critic networks

Initialize REDQ agent with critic parameters $\phi_1, \ldots, \phi_N$ and policy parameters $\theta$
Initialize target parameters $\theta' \leftarrow \theta$ and $\phi_i' \leftarrow \phi_i$, for $i = 1, \ldots, N$
Initialize replay buffer $\mathcal{R}$ with offline data $\mathcal{D}$
**for** $k = 0$ **to** $K$ **do**
    Sample mini-batch $\mathcal{B}$ of B transitions $(\boldsymbol{s}, \boldsymbol{a}, r, \boldsymbol{s}')$ from $\mathcal{R}$
    Update critic parameters $\phi_1, \ldots, \phi_N$ using Equation 3
    Update actor parameters $\theta$ using Equation 4 with $\alpha = \alpha_{\text{offline}}$
    Update target networks $\theta' \leftarrow \tau\theta + (1 - \tau)\theta'$ and $\phi_i' \leftarrow \tau\phi_i + (1 - \tau)\phi_i'$
**end for**

Randomly remove 95% of offline samples from $\mathcal{R}$
Initialize $\alpha_{\text{online}} = \alpha_{\text{offline}}$
Initialize $R_{\text{current}}$ and $R_{\text{last}}$ to store the return of current and previous episodes
Initialize environment for online fine-tuning
**for** every training episode **do**
    **for** $t = 0$ **to** $T$ **do**
        Act with exploration noise $\boldsymbol{a}_t \sim \pi_\theta(\boldsymbol{s}_t) + \mathcal{N}(0, \sigma_{\text{expl}})$
        Observe next state $\boldsymbol{s}_{t+1}$ and reward $r_t$
        Add $(\boldsymbol{s}_t, \boldsymbol{a}_t, r_t, \boldsymbol{s}_{t+1})$ to $\mathcal{R}$
        **for** $g = 0$ **to** $G$ **do**
            Sample mini-batch $\mathcal{B}$ of B transitions $(\boldsymbol{s}, \boldsymbol{a}, r, \boldsymbol{s}')$ from $\mathcal{R}$
            Update critic parameters $\phi_1, \ldots, \phi_N$ using Equation 3
            Update actor parameters $\theta$ using Equation 4 with $\alpha = \alpha_{\text{online}}$
            Update target networks $\theta' \leftarrow \tau\theta + (1 - \tau)\theta'$ and $\phi_i' \leftarrow \tau\phi_i + (1 - \tau)\phi_i'$
        **end for**
    **end for**
    Set $R_{\text{last}} = R_{\text{current}}$ and $R_{\text{current}} = \sum_{t=0}^{T} r_t$
    Adapt $\alpha_{\text{online}}$ based on $R_{\text{last}}$ and $R_{\text{current}}$ using Equation 2
**end for**

---

We combine this REDQ policy update with a behaviour cloning loss (like in Equation 1) for robust learning (Fujimoto & Gu, 2021):

$$\nabla_\theta \frac{1}{|B|} \sum_{(\boldsymbol{s}, \boldsymbol{a}) \in \mathcal{B}} \frac{1}{N} \sum_{i=1}^{N} \bar{Q}_{\phi_i}(\boldsymbol{s}, \pi_\theta(\boldsymbol{s})) - \alpha(\pi_\theta(\boldsymbol{s}) - \boldsymbol{a})^2. \tag{4}$$

We show that this simple modification of ensembling the critic networks (which can be run in parallel) improves offline-to-online learning. We call this algorithm REDQ+AdaptiveBC. Our algorithm is outlined as Algorithm 1.

## 5 EXPERIMENTS

### 5.1 ONLINE FINE-TUNING ON D4RL BENCHMARK

The goal of our experiments is to evaluate the stability and sample-efficiency of the proposed algorithm on online fine-tuning after offline pre-training on datasets of different quality. We evaluate our algorithm on online fine-tuning after offline pre-training on the D4RL benchmark (Fu et al., 2020). D4RL includes three locomotion tasks (halfcheetah, hopper, and walker) implemented in the MuJoCo simulator (Todorov et al., 2012), wrapped in OpenAI Gym API (Brockman et al., 2016). D4RL provides five different offline datasets for each task: Random, Medium, Medium-Replay, Medium-Expert, and Expert. The Random datasets are collected by random policies, Medium datasets are collected by an early-stopped soft actor-critic (SAC) (Haarnoja et al., 2018) agent with medium-level performance, Medium-Replay datasets consist of all samples in the replay buffer after training a medium-level agent, Medium-Expert datasets are mixed with expert demonstrations and sub-optimal demonstrations from a medium-level agent, and Expert datasets are expert demonstrations. The "expert" in these datasets is a fully trained soft-actor critic agent. We ignore the Expert

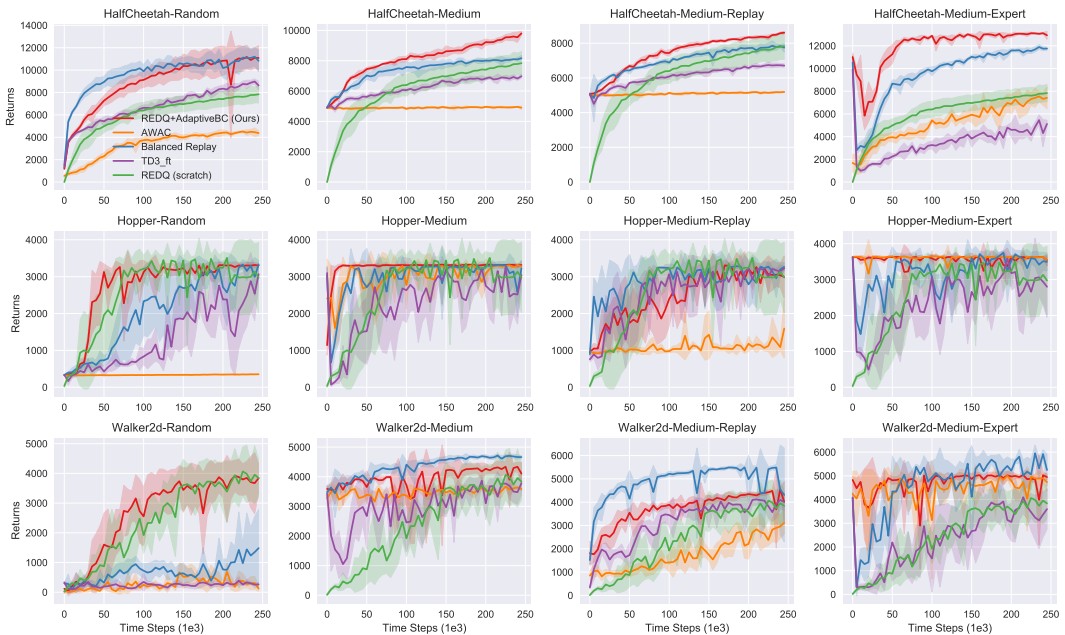

Figure 2: Results of online fine-tuning on the D4RL benchmark. We plot the mean and standard deviation across 5 runs. Our REDQ+AdaptiveBC method attains performance competitive to the state-of-the-art. It should be noticed that, unlike other methods, our results do not collapse immediately at the beginning of training.

datasets in this paper as offline RL algorithms already achieve expert-level performance on these tasks and there is little to no benefit in online fine-tuning.

In Figure 2, we compare our REDQ+AdaptiveBC algorithm with two state-of-the-art offline-to-online RL algorithms (AWAC and Balanced Replay) and two baseline methods (TD3-ft and REDQ):

- Advantage Weighted Actor-Critic (AWAC) (Nair et al., 2020) is an actor-critic method for offline-to-online RL that implicitly constraints the policy network to stay close to the behavior policy. We produce the results for AWAC using code taken from `https://github.com/ikostrikov/jaxrl`.

- Balanced Replay (Lee et al., 2021) is an offline-to-online RL method that prioritizes near-on-policy samples from the replay buffer. This method also uses an ensemble of Q functions to prevent overestimation of Q values in the initial stages of online fine-tuning. We reproduced the results for this method using our own implementation. For a fair comparison, we base our implementation on TD3+BC (instead of CQL originally used by Lee et al. (2021)) while ensuring that we are able to reproduce the original results.

- TD3-ft is the standard TD3 algorithm (Fujimoto et al., 2018) that was pre-trained offline using TD3+BC (Fujimoto & Gu, 2021).

- REDQ (scratch) (Chen et al., 2021) is an RL method trained from scratch, without any access to the offline data. This baseline emphasizes the importance of offline pre-training and online fine-tuning. We base our REDQ implementation on TD3 (instead of SAC used by Chen et al. (2021)) for compatibility with TD3+BC.

All methods (except AWAC) are implemented on top on TD3 and are run from the same codebase for a fair comparison. For simplicity, we do not perform any state normalization like in the original TD3+BC implementation (Fujimoto & Gu, 2021).

During offline pre-training, all algorithms are pre-trained on the offline dataset for one million gradient steps. After pre-training, we fine-tune the agents for 250,000 time steps by interacting with the environment. We evaluate the agent every 5000 time steps and each evaluation consists of 10

episodes. We attain performance competitive to the state-of-the-art in this benchmark with our method stably improving the performance during online fine-tuning.

We significantly outperform all others methods in the HalfCheetah domain for three datasets. We perform slightly worse than Balanced Replay on Walker2d-Medium and Walker2d-Medium-Replay but outperform it or get similar results in all other cases. We need to mention that in both Walker2d-Medium and Walker2d-Medium-Replay tasks, our method already reaches the target performance (predefined following D4RL), and the $\alpha$ is increased automatically to maintain the performance and thus fail to improve further. We significantly outperform REDQ on all tasks, which demonstrates that we considerably benefit from offline pre-training. TD3-ft is able to improve from online fine-tuning but suffers from significant performance drops due to the sudden distribution shift and the learning progress is slow due to the replay buffer being dominated by offline samples. It should be noticed that, unlike other methods, our algorithm does not collapse immediately on all three Medium-Expert tasks.

Both Balanced Replay and our method (REDQ+AdaptiveBC) use an ensemble of 10 Q networks, but in different ways. Balanced Replay maintains a pair of five ensemble networks, average the predictions across each of the five networks and then takes the minimum of the averages as the final prediction. In our method, we simply consider the average of all 10 networks as the prediction but randomly sample a pair of Q networks to compute the critic targets (Equation 3). We show that this simple modification enables stable and sample-efficient online fine-tuning without the need for any complex sampling scheme from the replay buffer.

Similar to prior works (Fujimoto & Gu, 2021), we use feed-forward networks with two hidden layers as actor and critic networks for all the methods. We use a batch size of 256 to train the network for all methods, except for AWAC where we use a larger batch size of 1024 (Nair et al., 2020). During offline learning, we use $\alpha_{\text{offline}} = 0.4$ for all tasks, except Walker-Random where we use $\alpha_{\text{offline}} = 100$ since the dataset has a very narrow distribution. We list all the hyperparameters used in our experiments in Table 1 in the Appendix.

## 5.2 ALGORITHMIC INVESTIGATIONS

**Adaptive Weighing of** $\alpha_{\text{online}}$: To evaluate whether the proposed method can correctly select a good $\alpha_{\text{online}}$ for stable online fine-tuning, in Figure 3, we compare the results obtained with the automatically tuned $\alpha_{\text{online}}$ with manually tuned results. To manually tune the $\alpha_{\text{online}}$, for each domain and each dataset, we do a grid search on $\alpha_{\text{online}}$ over $[0.0, 0.1, 0.2, 0.3]$ and pick the best $\alpha_{\text{online}}$ separately for each task.

We can see that with manually tuned $\alpha_{\text{online}}$, our method consistently outperformances other methods in Figure 2, except the Balanced Replay on Walker2d-Medium. Our results with automatically tuned $\alpha_{\text{online}}$ are slightly worse than manually tuned results on HalfCheetah tasks. However, our method successfully find the similar $\alpha_{\text{online}}$ on Halfcheetah tasks as we manually selected after roughly 10-15 episodes. On the rest tasks, our automatically tuned results are competitive to the carefully picked results but saving lots of labor and computational resources.

**Offline Dataset Downsampling**: Balanced Replay (Lee et al., 2021) trains a neural network to estimate the priority of samples from offline data and online data. In their methods, three replay buffer need to be maintained: offline dataset (0.1-2M samples), online dataset (0.25M samples) and a prioritized replay buffer (0.35M-2.25M samples) (Schaul et al., 2015), making it memory consuming (0.7M-4.5M samples). Our method simply dowsamples the offline dataset by $95\%$, thus, our method only maintains one replay buffer to store online data but is prefilled with 0.05M offline data points, roughly saves $65\% - 95\%$ memory.

To demonstrate the effectiveness of dataset downsampling, we compare TD3_ft with and without dataset downsampling on three random datasets, shown in Figure 4. Our results show that the downsampling procedure allows the agent effectively sampling the novel data encountered during fine-tuning. This is even important when the data quality of the offline data is not good enough, such as when the dataset is collected by a random policy.

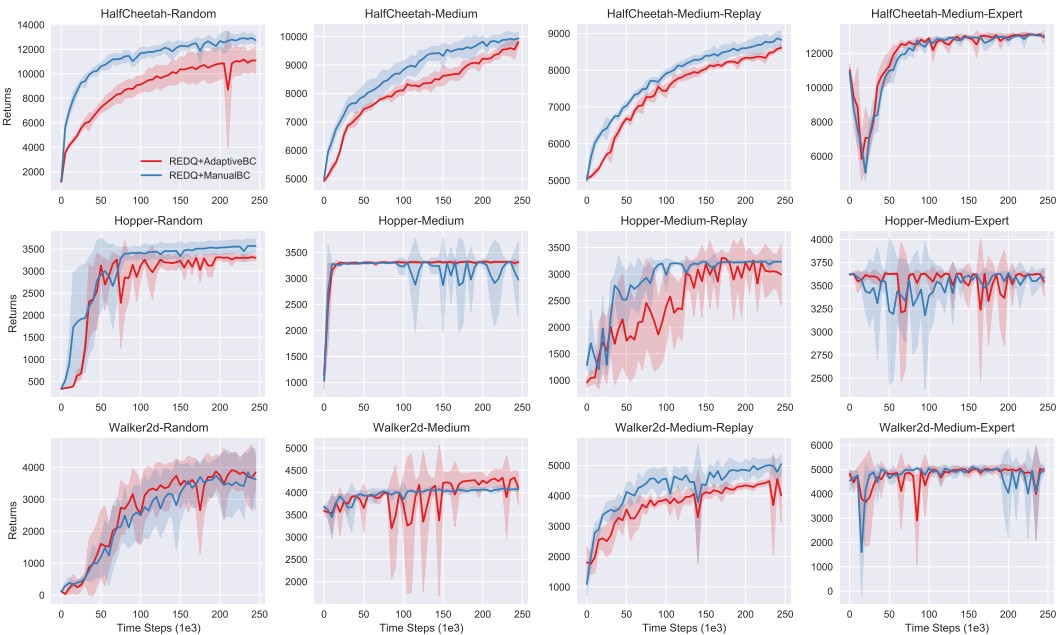

Figure 3: Comparison of results with automatically tuned $\alpha_{\text{online}}$ and carefully picked results. It shows that our proposed method can effectively find the suitable $\alpha_{\text{online}}$ for all tasks.

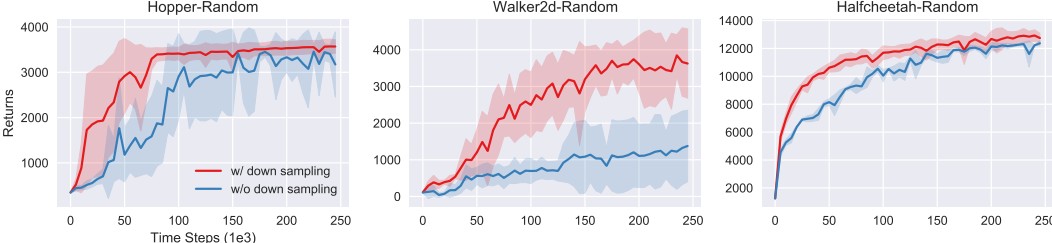

Figure 4: Comparison of TD3-ft with and without dataset downsampling. We plot the mean and standard deviation across 3 runs. Downsampling enables effective usage of novel data encountered during fine-tuning.

## 6 CONCLUSION

We consider the problem of offline-to-online RL where an agent is first pre-trained on offline data collected (by a possibly unknown behavior policy) and the agent is then fine-tuned online by interacting with the environment. This is desirable as pre-trained agents may have limited performance depending on the quality of the offline dataset. Offline-to-online RL is challenging due to the sudden distribution shift from offline data to online data, and also the constraints enforced by offline RL algorithms (such as a behavior cloning loss) during pre-training. In this paper, we propose a simple mechanism to adaptively weight a behavior cloning loss during online fine-tuning, based on agent performance and training stability. We demonstrate that a randomized ensemble further helps to deal with these challenges to enable sample-efficient online fine-tuning performance. We attain performance competitive to the state-of-the-art online fine-tuning methods on locomotion tasks from the popular D4RL benchmark.

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

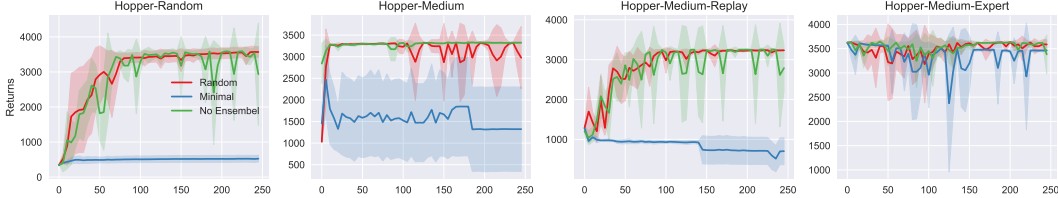

Figure 5: Comparison of usages of ensembles on the hopper domain. We plot the mean and standard deviation across 3 runs. Randomized ensembled double Q-Learning stabilizes the training, but it is not necessary to avoid performance collapse during fine-tuning.

Table 1: Hyperparameters used in our experiments

|  | Hyperparameter | Value |
|---|---|---|
| TD3 | Optimizer | Adam |
|  | Learning rate | 3e-4 |
|  | Batch size | 256 |
|  | Target update rate | 5e-3 |
|  | Policy noise std | 0.1 |
|  | Policy noise clip | 0.5 |
|  | Policy update frequency | 2 |
| Architecture | Hidden layers | 2 |
|  | Hidden units | 256 |
|  | Activation function | ReLU |
| REDQ | Number of networks N | 10 |
|  | Randomly sampled networks M | 2 |
|  | Number of updates G | 20 |
| Offline BC | $\alpha_{\text{offline}}$ | 0.4 |
| Adaptive BC | $K_p$ | 3e-5 |
|  | $K_d$ | 8e-5 |

## A    ABLATION ON ENSEMBLES

In our experiments, we use ensembles to represent the critic network. In 5, we compare the training results with and without ensembles as well as the way to use ensembles. Our results show that using ensembles is not necessary to avoid performance collapse, however, it stabilizes training in most cases. Also, the way to use ensembles matters. We compare calculating target Q values with randomly sampled Q predictions and with minimal predictions. Our results show that using minimal Q predictions to calculate target Q values hurts the performance in most cases.

## B    HYPERPARAMETERS

All the hyperparameters and network architectures used in experiments are listed in Table 1.

