# OpenReview forum: "Adaptive Behavior Cloning Regularization for Stable Offline-to-Online Reinforcement Learning"
_ICLR.cc/2022/Conference — ICLR 2022 Submitted_

### Official Review · Reviewer_bnwi · 2021-10-17

**Correctness:** 4
**Technical Novelty And Significance:** 2
**Empirical Novelty And Significance:** 3
**Recommendation:** 5
**Confidence:** 3

**Main Review:**

The paper is well organized and the motivation is really clear. Offline-to-online learning considered in this paper is an important problem. I agree that adaptively updating the behavior policy constraint is the key point in online fine-tuning. In the experiments, the authors compare the two SOTA offline-to-online methods AWAC and Balanced Replay, and ablation studies show the effectiveness of adaptive $\alpha$.

My main concern is the novelty. REDQ and TD3+BC are prior work, and it seems that the main contribution in this paper is the adaptive mechanism of $\alpha$. Although this paper tries to solve an important problem, the contribution of the proposed method is limited.

The update rule of $\alpha$ is heuristic. Although it is well-motivated, more theoretical analysis is expected to demonstrate how the adaptive  $\alpha$ achieves the balance between performance improvement and performance dropping. And the selection of hyperparameters KP and KD is arbitrary. More empirical results are needed to show how the two hyperparameters affect the update of $\alpha$​. In Figure 3, AdaptiveBC does not significantly outperform ManualBC. Although ManualBC uses grid search, the search space is not too large.

In Figure 2, if no expert data is involved, REDQ (scratch) could achieve a good performance in 250000 timesteps, which shows pure online learning is enough to deal with this setting. In that case, it is important to consider the sample efficiency and deployment efficiency[1] in offline-to-online learning.

[1]https://arxiv.org/abs/2006.03647

**Summary Of The Paper:**

The authors combine REDQ and TD3+BC and propose an adaptive mechanism to update $\alpha$ during the online fine-tuning. The proposed method decreases $\alpha$ for further improvement and increases  $\alpha$  to avoid performance collapse.

**Summary Of The Review:**

The paper aims to solve an important problem: adaptive behavior policy constraint in offline-to-online learning. However, the proposed method is heuristic, and more theoretical analysis is expected.

---

> ### Author Response · Authors · 2021-11-23
> **Response to Reviewer bnwi**
>
> We thank you for the review. Please see the common response for a response to the main concerns. We now address the remaining concerns:
>
> 1. “In Figure 3, AdaptiveBC does not significantly outperform ManualBC. Although ManualBC uses grid search, the search space is not too large.“
>
> ManualBC is not possible in practice. In a real-world problem, after offline pre-training, we can only perform online fine-tuning once. It is not possible to perform a grid search of the alpha hyperparameter during this online fine-tuning as it is highly inefficient and even unsafe for some applications. We show that AdaptiveBC can automatically tune the alpha hyperparameter for any task, during online fine-tuning. AdaptiveBC is able to perform almost as well as ManualBC without any manual tuning.
>
> 2. Sample efficiency and deployment efficiency in offline-to-online learning
>
> We agree that sample efficiency and deployment efficiency are important problems to consider during offline-to-online learning. We study the problem of offline-to-online RL which can be applied in cases where the offline data is beneficial and online fine-tuning can be used to further improve the performance in a sample-efficient manner.

---

### Official Review · Reviewer_aXQe · 2021-10-29

**Correctness:** 3
**Technical Novelty And Significance:** 2
**Empirical Novelty And Significance:** 2
**Recommendation:** 3
**Confidence:** 5

**Main Review:**

## Strengths
- This paper is well-written and easy to follow.
- The focus of this paper, fine-tuning in RL seems to be a significant topic towards real-world applications, overcoming the bottleneck of current offline RL research.
- The empirical comparisons seem to show better or competitive performance to AWAC, Balanced Replay, TD3_ft, and REDQ.

## Weaknesses
- The proposed method REDQ+AdaptiveBC seems to require a lot of hyper-parameter tuning. First, $\alpha_{offline}=0.4$ is a very small value compared to original TD3+BC, which uses $\alpha_{offline}=2.5$ after a grid search from \{1.0, 2.0, 2.5, 3.0, 4.0\}. This implies that $\alpha_{offline}=0.4$ (and 100 for walker2d-random) might be a result of extensive hyper-parameter tuning. In addition, I guess $K_p$, $K_d$ (and even target return) must be well-balanced. I'm not sure how they are determined.
- Also, REDQ+AdaptiveBC seems incremental on top of TD3+BC, combining the established Q-ensemble method (Chen et al. 2021) and carefully-designed coefficient tuning, and resembles Balanced Replay (Lee et al. 2021). I summarized the problems in fine-tuning and the remedies for them on the table. Both Balanced Replay and REDQ+AdaptiveBC overcome the data distribution shift issues using the ensembles of Q-functions. For imbalance issues of the offline-online replay buffer, Balanced Replay utilizes  Prioritized Experience Replay, and REDQ +AdaptiveBC downsamples the offline data, both of which aim to reduce the ratio of offline data and encourage to leverage the online samples when online fine-turning. During the online training phase, behavior regularization for stable offline training can prevent the improvement of the policy. Balanced Replay, proposed on top of CQL, removes the CQL loss and train value function and policy as online SAC, and REDQ+AdaptiveBC gradually reduce the ratio of BC loss. I think the novelty and contribution of this paper are not enough.

|Problem|Balanced Replay (Lee et al. 2021) | REDQ+AdaptiveBC  |
| ---- | ---- | ---- |
|  **Distribution Shift**  |  Pessimistic Q-Ensemble  |  REDQ (Chen et al. 2021)  |
|  **Imbalanced Replay Buffer** |  PER  |  Downsampling  |
|  **Behavior Regularization**  | CQL-SAC switching |  Adaptive $\alpha_{online}$  |

- I think the comparison might be insufficient, since Balanced Replay, proposed on top of CQL, is based on TD3+BC in this paper. I don't think the comparison to CQL-based Balanced Replay isn't fair. In addition, while adaptive BC heavily depends on TD3+BC algorithms, Balanced Replay seems to work on both CQL and TD3+BC. I also think the authors can add TD3+BC baseline (keeping $\alpha_{offline}$ during online training) and +REDQ/-REDQ ablations.

### Minor Comments
- I'm not sure how can I interpret the results of Figure 4. In my understanding, downsampling of replay buffer is introduced to overcome the instability of the beginning of the online training due to the imbalanced replay buffer (as discussed in Lee et al. 2021), but the caption says "Downsampling does not hurt the training". Doesn't downsampling improve the performance?
- You can consider adding Ghasemipour et al. (2021), that proposes the ensemble-based offline RL algorithms, to the related work.
- Prior works of offline-to-online fine-tuning (AWAC, Balanced Replay) work on robotic environments. Comparison on these environments, not only current MuJoCo locomotion, might be helpful.


### Reference
Fujimoto and Gu. A Minimalist Approach to Offline Reinforcement Learning.  ArXiv preprint arXiv:2106.06860 (2021).

Lee et al. Offline-to-Online Reinforcement Learning via Balanced Replay and Pessimistic Q-Ensemble. Conference on Robot Learning (2021).

Chen et al. Randomized Ensembled Double Q-Learning: Learning Fast Without a Model. International Conference on Learning Representations (2021).

Ghasemipour et al. EMaQ: Expected-Max Q-Learning Operator for Simple Yet Effective Offline and Online RL. International Conference on Machine Learning (2021).


**Summary Of The Paper:**

This paper studies the fine-tuning problem from offline to online RL. While the naive approach to fine-tune offline policy suffers from a sudden distributional shift by online samples and too much behavior constraint in offline algorithms. The proposed method leverage (1) the adaptive coefficient tuning in TD3+BC loss, (2) randomized ensembles of Q-functions (proposed by Chen et al. 2021), and (3) down-sampling of offline data. The experiments seem to show better or competitive results to other approaches.

**Summary Of The Review:**

As I described in Main Review, this paper seems to require extensive hyper-parameter search (proper $\alpha_{offline}$ and well-balanced $K_p$, $K_d$), and to propose the incremental algorithms on top of TD3+BC (and resemble Lee et al. 2021), combining Q-ensemble method proposed by Chen et al. (2021), downsampling for imbalanced replay buffer, and adaptive $\alpha_{online}$ tuning. Due to the lack of novelty and significant contributions to the ICLR (or RL) community, I vote for rejection.

---

> ### Author Response · Authors · 2021-11-23
> **Response to Reviewer aXQe**
>
> We thank you for the review. Please see the common response for a response to the main concerns. We thank you for pointing out Ghasemipour et al. (2021). We will add it to the related work. We now address the remaining concerns:
>
> 1. “$\alpha_{offline}$ is a very small value compared to the original TD3_BC, which uses 2.5”
>
> We do not tune this hyperparameter. Note that we use the alpha hyperparameter to weigh the BC loss term while in the TD3_BC paper, the alpha hyperparameter was used to weight the Q maximization term. So, we simply set $\alpha_{offline} = 1 / 2.5 = 0.4$.
>
> 2. “requires a lot of hyper-parameter tuning”
>
> As we point out above, we do not tune the alpha hyperparameter. We tune the K_p and K_d hyperparameters using grid search but they are the *same* for all tasks.
>
> 3. Comparison to Balanced Replay
>
> We reimplemented Balanced Replay by changing the offline algorithm from CQL to TD3_BC, to ensure a fair comparison. We also ensured that the reproduced results of Balanced Replay are comparable to the results reported by the authors.
>
> In comparison to Balanced Replay, we would like to stress the *simplicity* of our method. Balanced Replay needs to train another network to assign priority to data points. In order to do this, they also need to keep three replay buffers, while our method shows that downsampling together with adaptive alpha is enough to avoid performance collapse. Also, in three medium-expert tasks, the performance of Balanced Replay drops immediately during fine-tuning, while our method successfully avoids the performance collapse.
>
> 4. "Downsampling does not hurt the training" in Figure 4
>
> We thank you for pointing this out. We have updated the caption to clarify that “downsampling enables effective usage of novel data encountered during fine-tuning”.

---

> > ### Comment · Reviewer_aXQe · 2021-11-24
> > **Re: Response to Reviewer aXQe**
> >
> > I appreciate the author to respond the review.
> >
> > **> comment to 1**
> >
> > I found that the proposed objective is derived by dividing the objective of TD3+BC by $\alpha$. So $\alpha=0.4$ seems decent choice.
> > After I read the response, I wonder why $\alpha=100$ is used for walker2d-random. The authors say "since the dataset has a very narrow distribution." While REDQ (scratch) can learn the good policy without BC term, does REDQ+Adaptive BC require a large weight for BC term? (I found that $\alpha_{offline}$ is the initial value of $\alpha_{online}$. Large $\alpha_{online}$ is expected  if $\alpha_{offline}$ is given (i.e. BC is preferred than RL)). Can the authors provide any intuitions?
> >
> > **> comment to 2**
> >
> > I think the range of $K_p$ and $K_d$ for grid search and its summary table or figures (e.g. Appendix A.3 in https://arxiv.org/abs/1911.11361, which covers the wide range of values) of the results would be helpful to show the robustness to the hyper-parameters.
> >
> > **> comment to 3**
> >
> > If the authors emphasize the simplification of replay buffer as a contribution, I'd like to see further detailed analysis. The authors provide the dataset downsampling ablations on the random dataset only, but mention the medium-expert tasks as examples where the proposed method is superior to balanced replay. The ablation of what kind of data should be kept, and that of the difference from the data that Balanced replay kept might provide more important insights.
> >
> > **> comment to shared 3. Evaluation on more challenging tasks**
> >
> > As the reviewer q4CJ pointed out, offline-to-online works (AWAC, Balanced Replay) often work on robotic tasks, and they are not so high-dimensional tasks (except for image-based tasks). For example, Balanced Replay is tested on robotic manipulation (https://github.com/avisingh599/roboverse), which has 3-dim object location, 4-dim object rotation, and 10-dim robot state. Totally 17-dim observation is comparable to halfcheetah or walker2d tasks. Since offline-to-online RL is expected to the application to the real world, I think the evaluation on such realistic tasks would be valued.

---

> > > ### Author Response · Authors · 2021-12-03
> > > **Response to Reviewer aXQe**
> > >
> > > 1. “why α=100 is used for walker2d-random”
> > > The reason to use larger walker2d-random is that the offline dataset of walker2d-random has a very narrow distribution, and most of the current offline RL methods (TD3_BC, CQL, etc) fail to prevent the Q function from diverging. We found that after 1M steps of offline training, the Q value grows greater than 1e+9 when using the default alpha. To avoid this erroneous starting point for online fine-tuning, we increase the alpha such that the Q value stays at a reasonable value after pre-training.
> > >
> > > 2. “While REDQ (scratch) can learn the good policy without BC term, does REDQ+Adaptive BC require a large weight for BC term?”
> > > While REDQ from scratch can learn a good policy, the BC term is required so that there is no performance collapse when we switch from offline to online training. Actually on all three *random* datasets, within 20K timesteps, the alpha_online will reduce to 0, which aligns with REDQ without the BC loss.
> > > We keep the initial alpha_online as the same (0.4). This is because when receiving new data, the Q function won’t dramatically diverge as in offline RL.
> > >
> > >  3. “The ablation of what kind of data should be kept, and that of the difference from the data that Balanced replay kept might provide more important insights.”
> > > We add more ablation studies on dataset downsampling. We compare two sampling methods (retain randomly sampled data or retain trajectories with high episodic returns) as well as different sampling ratios [0.05, 0.1, 0.5, 1.0]. We find that retaining all offline dataset (ratio=1.0) hurts the fine-tuning training, even on Medium-Expert datasets. A lower downsampling ratio (ratio=0.05) encourages the efficient usage of the novel data encountered during fine-tuning.
> > >
> > > 4. “Evaluation on more challenging tasks”
> > > Thanks for pointing out, we agree that more realistic tasks will be valuable, thus, based on the D4RL benchmark, we evaluate our algorithm on the Expert dataset of the four dexterous manipulation tasks (Door, Hammer, Relocate, Pen), each composed of one million expert data from a fine-tuned RL policy. We first tune TD3\_BC for offline pre-training on these tasks. We increase the $\alpha_{offline}$ from 0.4 to 8, and correspondingly increase the initial $\alpha_{online}$, $K_p$, $K_d$ by the same factor of 20. We observe that the performance of the REDQ agent immediately collapses but the proposed adaptive behavior cloning method is able to successfully prevent this.

---

### Official Review · Reviewer_q4CJ · 2021-11-02

**Correctness:** 4
**Technical Novelty And Significance:** 3
**Empirical Novelty And Significance:** 2
**Recommendation:** 5
**Confidence:** 5

**Main Review:**

I think the paper is clearly written and easy to understand. The proposed algorithm is also novel in the sense that it gives an adaptive scheme of selecting the amount of conservatism during online training depending on the current policy performance. The empirical results also clearly show that the method can prevent the policy from degrading in medium-expert datasets where we don't need to change the current policy much, aligning with the initial motivation of the method.

I also have a few concerns. First, while the results of the method generally align with the intuition that the policy should use large alpha on narrow datasets with good data and use large alpha on diverse datasets with low-quality data, the performance of the method on hopper and walker2d is a bit underwhelming. The method seems to be the same as hopper and also worse than Balanced Replay on three of the walker datasets. While the authors mentioned that the less satisfying results on walker2d are caused by the predefined target score in D4RL, I wonder if that is potentially a limitation of the method since it relies on the human-defined target score. Also, how would the method do if we change the target score to be higher (e.g. R_max * T)?

Moreover, I think the paper lacks some theoretical understanding of the method. The intuition of the method makes perfect sense, but I would like to see how this adaptive scheme can give use some theoretical insights on policy improvement guarantees. Prior works [1] have studied this and I wonder how the proposed method can be connected to that.

Finally, beyond the mujoco tasks, I think it would be good to evaluate the method on more realistic tasks such as the dexterous manipulation tasks used in the AWAC paper.

Minor comment: it would be great to have an ablation study on the REDQ ensemble since that seems to be a bit disjoint to the theme of the paper.

[1] Xie, Tengyang, et al. "Policy Finetuning: Bridging Sample-Efficient Offline and Online Reinforcement Learning." arXiv preprint arXiv:2106.04895 (2021).

**Summary Of The Paper:**

This paper proposes a new offline RL with online fine-tuning method. The authors first pretrain the policy using recent offline RL method TD3+BC with offline data and then collect on-policy data to further improve the pretrained policy. To prevent the policy from either degrading performance or failing to improve at the fine-tuning stage, the authors propose an automatic scheme to adjust the $\alpha$ term that controls the BC loss in TD3+BC to ensure that the policy does not continue to contain itself to the behavior policy if there's room for improvement and also is able to stick to the previous policy if the performance is already near optimal. The authors conduct evalutions in D4RL mujoco environments and show that the approach is able to outperform prior methods in halfcheetah.

**Summary Of The Review:**

Given my comments in the above section, I think the paper could be improved if the authors can show the results of the method with a more realistic target score (e.g. R_max * T), provide the theoretical justifications, and evaluate the method on more realistic domains. I would vote for a weak reject given the current status.

---

> ### Author Response · Authors · 2021-11-23
> **Response to Reviewer q4CJ**
>
> We thank you for the review. Please see the common response for a response to the main concerns.
>
> 1. “This method seems to be the same as hopper and also worse than Balanced Replay on three of the walker datasets”
>
> We would like to point out that the performance of our method *doesn’t drop dramatically on expert-medium tasks* like Balanced Replay. Also, our method doesn’t need to train an additional network to assign priority to sampled data like Balanced Replay, which saves *65%-95%* memory usage (as we describe in Section 5.2). Our method can be potentially combined with the Balanced Replay method to improve sample efficiency.
>
> 2. “Prior works [1] have studied this and I wonder how the proposed method can be connected to that.”
>
> The reference [1] sounds very relevant but is missing from the review. Can you share that?

---

> > ### Comment · Reviewer_q4CJ · 2021-11-24
> > **Thank you for the reply; Some questions and comments**
> >
> > Thank you for replying to my comments! See below for my detailed response to several points raised in your reply.
> >
> > 1. Regarding the performance of the method on D4RL tasks and memory efficiency, I agree that the method can prevent the performance from dropping drastically on medium-expert tasks and also is memory-efficient. However, I think in other settings such as medium and medium-replay, the method is worse than or similar to Balanced Replay especially on hopper and walker2d tasks. Therefore, it seems that the improvement of the method over prior methods appears a bit limited. Moreover, I'm not sure if memory efficiency is that much a benefit especially in low-dimensional settings such as D4RL.
> >
> > 2. "The reference [1] sounds very relevant but is missing from the review. Can you share that?" Sorry about this. By [1], I'm referring to this paper (Xie, Tengyang, et al. "Policy Finetuning: Bridging Sample-Efficient Offline and Online Reinforcement Learning." arXiv preprint arXiv:2106.04895 (2021).).
> >
> > 3. Regarding the human-defined score, I think it's realistic in some settings but generally a less practical assumption since for many complex tasks, it's hard to get an expert score. I'm also not sure if the naive R_max * T target score will work well since in that case, the method will always try to be less conservative and might still suffer from the performance drop. Empirical results on this would be important.
> >
> > 4. Regarding evaluation on more challenging tasks, I agree that trying high-dimensional image-based tasks is challenging for offline RL, but I'm suggesting that you could try the dexterous manipulation tasks (i.e. adroit tasks) in the AWAC paper, which are also low-dimensional but are more realistic.

---

> > > ### Author Response · Authors · 2021-12-03
> > > **Response to Reviewer q4CJ**
> > >
> > > 1. “ in other settings such as medium and medium-replay, the method is worse than or similar to Balanced Replay especially on hopper and walker2d tasks ”
> > >
> > > Since the submission, we have fine-tuned the hyperparameters of the PID controller and now on halfcheetah-medium, halfcheetah-medium-replay, walker2d-medium, walker2d-medium-replay, and hopper-medium-replay tasks, our AdaptiveBC method matches or outperforms Balanced Replay. On the hopper-medium task, our method is slightly worse than the Balanced Replay method at the beginning, but after 100k timesteps, it roughly reaches the same performance. Also, the use of prioritized replay (instead of the simple downsampling that we do) is orthogonal to our method, so the results could potentially be improved by combining them.
> > >
> > > 2. “I'm not sure if memory efficiency is that much a benefit especially in low-dimensional settings such as D4RL”.
> > >
> > > Memory efficiency is one benefit of our method, and a simple yet efficient method is always preferred. We show that by simply downloading the dataset, combining it with the adaptive bc term, we get better or comparable performance on all tasks among different environments and different datasets. In the Balanced Replay paper, a neural network is needed to estimate the density ratio of new data and low data, which is already a challenging task, especially when inputs are raw images. Furthermore, with the development of offline RL as well as offline-to-online RL, memory efficiency should be appreciated since we need to consider high-dimensional inputs (e.g. images) as well as large-scale datasets in real-world applications.
> > >
> > > 3. The recommended paper [1] is highly related to our work, we will include it in our related work. Thanks for the recommendation.
> > >
> > > 4. “I'm also not sure if the naive R_max * T target score will work well since in that case, the method will always try to be less conservative and might still suffer from the performance drop. Empirical results on this would be important..”
> > >
> > > We estimate R_max as the highest reward attained by an expert policy and then set R_max*T as the target return for PID control. We do not perform any additional hyperparameter tuning. While using R_max*T as the target, the learning curves are very similar for 11 out of 12 D4RL locomotion tasks. On hopper-medium-replay, the performance is slightly unstable in the beginning but the performance quickly recovers, to outperform other methods.
> > >
> > > 5. “try the dexterous manipulation tasks (i.e. adroit tasks) in the AWAC paper”
> > >
> > > Thanks for the suggestion. We agree that more realistic tasks will be valuable. Thus, based on the D4RL benchmark, we evaluate our algorithm on the Expert dataset of the four dexterous manipulation tasks (Door, Hammer, Relocate, Pen), each composed of one million expert data from a fine-tuned RL policy. We first tune TD3\_BC for offline pre-training on these tasks. We increase the $\alpha_{offline}$ from 0.4 to 8, and correspondingly increase the initial $\alpha_{online}$, $K_p$, $K_d$ by the same factor of 20. We observe that the performance of the REDQ agent immediately collapses but the proposed adaptive behavior cloning method is able to successfully prevent this.

---

### Official Review · Reviewer_6n3w · 2021-11-02

**Correctness:** 3
**Technical Novelty And Significance:** 2
**Empirical Novelty And Significance:** 2
**Recommendation:** 5
**Confidence:** 4

**Main Review:**

Pros:

** This paper considers an important problem setting for RL.

** Paper is well written and is easy to follow.

Cons:

** The proposed solution is heuristic and is similar to other methods that use annealing to tune the regularizer weight. And similar to other heuristics in this space, this method also has this limitation that it requires pre knowledge about the final performance of the task. I would like to see a more principled solution that at-least does not require such knowledge.

** Evaluations are done on simple low dimensional tasks. Given the limited technical contribution, i would like to see a study on more complicated tasks with for example visual inputs where representation learning introduce additional challenges for finetuning.

** The other claimed contribution on using ensemble of Q functions is orthogonal to the problem under study as it has shown before that ensemble of Q-functions helps to stabilize learning.

**Summary Of The Paper:**

This paper proposes an update rule to adaptively change the regularizer term (BC here) weight during online learning for more stable fine tuning to improve over an expert.

**Summary Of The Review:**

This paper considers an important topic for RL. However the proposed solution is heuristic and evaluations are not convincing given the simplicity of the tasks. Overall i think this paper does not offer enough contribution in its current form.

---

### Author Response · Authors · 2021-11-23
**Response to common concerns**

We thank the reviewers for the time and expertise invested in their constructive feedback. We are encouraged that all reviewers recognize that we study an important problem, the motivation of the paper is clear, and find the paper well-written and easy to read. We first address the common concerns of all reviewers:

1. Heuristic approach

We automatically adapt the alpha for each task using the PD controller. We control alpha so as to achieve a specific target performance. Intuitively, the alpha should be high for tasks where the policy learned from the offline dataset is already near-optimal and alpha should be low for tasks where the offline policy has to be significantly improved. We demonstrate that the PD controller is able to effectively control the alpha to achieve the target performance. While the PID controller has been studied extensively, we are working on performing more theoretical and empirical analyses of the proposed approach to gain theoretical insights on policy improvement guarantees.

2. Human-defined target score

While the human-defined target score in our approach can be a limitation in some cases, this can also be realistic in many MDPs where the cumulative rewards in an episode are bounded within a known range. For example, all the tasks defined in the DeepMind control suite have the cumulative rewards in an episode bounded within the range [0, 1000]. In our experiments on the D4RL benchmark, we use a target score of 1 as defined by the benchmark. We expect the proposed approach to work even with a target score that is higher than this. As Reviewer q4CJ suggests, for example, the target score could be calculated as R_max * T. We are evaluating the proposed approach with such a target score and will update the paper with the results.

3. Evaluation on more challenging tasks

Our work builds on top of the recent work on offline RL. We would like to point out that most of the research on offline RL is tested on low-dimensional tasks, such as the D4RL benchmarks tasks we use in the paper. Offline RL with high dimensional inputs is still a challenging problem and we are working on evaluating our proposed approach on higher-dimensional tasks.

4. Importance of ensemble of Q functions

The proposed approach works similarly even without an ensemble of Q functions. We have updated the paper with an ablation study (Figure 5) to show that while the ensemble of Q functions slightly improves the stability of online fine-tuning, it is not necessary for stable online fine-tuning.

We separately address the remaining concerns below.

---

> ### Comment · Reviewer_6n3w · 2021-11-30
> **RE: Response to common concerns**
>
> Thank you for the reply. After reading the other reviews and the rebuttal i think paper in its current form is not ready for a solid publication so i stay with my current score. In particular alleviating the mentioned limitations of the approach (to make a larger delta w.r.t. current existing heuristics) as well as better evaluation could improve the paper significantly. Thanks again.

---

### Decision · Program_Chairs · 2022-01-20

**Decision:**

Reject

**Comment:**

The paper proposes an approach that allows online finetuning of an offline RL policy by adaptively changing a BC regularization term.

Even after discussions with the authors, the reviewers had several concerns. First, the paper seems to be limited in novelty as the "REDQ+AdaptiveBC seems incremental on top of TD3+BC". Second, there were concerns that the adaptive regularization term was insufficient as a contribution given its heuristic nature.

Given the consensus among reviewers of this paper, I recommend rejecting this paper.